# Sparsity Distribution Matters: REACT for Accelerating Large Language Models

## Abstract

Efficient inference for large language models (LLMs) is critical for real-world deployment, yet it requires substantial computational and memory resources. Fortunately, activation sparsity alleviates these demands by enabling the skipping of low-magnitude activations, which reduces both arithmetic operations and memory access. However, existing methods primarily focus on maximizing the overall sparsity, but they overlook the impact of sparsity distribution in the inference network. Our empirical study with current methods reveals that sparsity distribution is more critical than the overall sparsity ratio for acceleration. Therefore, we propose REACT, a training-free sparsification method that optimizes sparsity distribution within the Multi-Layer Perceptron (MLP) module, improving inference speed without sacrificing model performance. Specifically, we empirically select the best location for sparsification in an MLP and develop an optimized sparsity-aware GPU kernel for inference, which reduces memory access overhead and improves computational efficiency. Our experiments on LLaMA2-7B and Mistral-7B demonstrate that REACT achieves speedups of 1.26× and 1.33×, respectively, while maintaining nearly the same model accuracy as their baselines. These results highlight the importance of rethinking sparsity distribution for efficient LLM inference.

## 1 Introduction

Large language models (LLMs) have demonstrated remarkable performance across various natural language processing (NLP) tasks, enabling significant advancements in machine translation (Achiam et al., 2023), text generation (Touvron et al., 2023), and code synthesis (Roziere et al., 2023). However, their inference cost remains a major bottleneck (Pope et al., 2023; Weng, 2023), particularly during auto-regressive decoding, where each token generation step involves passing the input through billions of parameters. As a result, optimizing LLM decoding efficiency has become a critical research direction, especially for latency-sensitive applications.

A promising direction for improving inference efficiency is activation sparsity (Liu et al., 2023), which leverages the observation that many activation values in Multi-Layer Perceptrons (MLPs) contribute little to the final output and can be safely pruned. Activation sparsity has been widely leveraged in ReLU-based LLMs (Mirzadeh et al., 2023), where the activation function itself enforces sparsity. However, in modern SiLU-based (Shazeer, 2020) architectures, activations are inherently dense, limiting the effectiveness of post-activation sparsity methods such as CATS (Lee et al., 2024), which applies pruning only to SiLU outputs and thus overlooks potential sparsity in other hidden states. This raises a critical question: Can we identify better sparsification targets within the MLP module to achieve higher sparsity while maintaining model performance?

To answer this question, we systematically investigate the sparsity properties of different hidden states within the MLP module. We find that applying sparsity after the up transformation in the MLP computation yields significantly better performance trade-offs. Compared to CATS, our method sparsifies a more effective location, achieving higher sparsity while maintaining comparable model accuracy. Moreover, while TEAL (Liu et al., 2024) sparsifies all MLP weight matrices uniformly, our findings indicate that the distribution of sparsity across MLP submodules is more important for acceleration than overall sparsity. TEAL achieves higher overall sparsity but results in suboptimal speedup due to inefficient sparsity distribution.

In this work, we propose REACT (**RE**thinking **ACT**ivation Sparsity for Faster LLM Decoding), a training-free method that extends activation sparsity to modern SiLU-based LLMs. Our method carefully selects sparsification targets within the MLP module, achieving higher sparsity than CATS while maintaining comparable model accuracy. Furthermore, our analysis demonstrates that sparsity distribution plays a critical role in acceleration, enabling REACT to translate MLP sparsity into actual speedup more effectively than TEAL. To efficiently exploit this sparsity, we develop a sparsity-aware custom GPU kernel, achieving up to 1.26x and 1.33× wall-clock speedup in LLaMA2-7B and Mistral-7B decoding, respectively.

The remainder of this paper is organized as follows: Section 2 reviews related work on activation sparsity and efficient LLM inference. Section 3 and 4 introduces the proposed REACT method. Section 5 presents extensive experiments validating our approach. Finally, Section 6 concludes with future directions.

## 2 RELATED WORK

To enhance the efficiency of large language model (LLM) inference, researchers have explored various sparse computation techniques (Hoefler et al., 2021). These methods can be broadly categorized into weight sparsity (pruning directly on weights) and activation sparsity (pruning on activations).

Weight pruning removes unimportant parameters from neural networks to reduce storage and computational cost while maintaining accuracy. Given the large scale of LLMs, pruning methods generally focus on post-training pruning to avoid the costly retraining process. These methods leverage gradient information (Ma et al., 2023) or weight magnitudes (Sun et al., 2023) to identify and prune redundant weights, enabling efficient training-free sparsification. However, to effectively translate weight pruning into wall-clock speedup, the resulting sparsity pattern must align with hardware-friendly structures (e.g., Nvidia's 2:4 sparsity pattern (Mishra et al., 2021)). This requirement imposes constraints on pruning patterns, limiting the flexibility of weight sparsity approaches. As a result, pruning the weights cannot fully exploit the acceleration potential of sparsity in LLM inference.

Activation sparsity (Mirzadeh et al., 2023; Song et al., 2024; Zhang et al., 2024) provides an alternative approach by selectively pruning activations rather than weights, reducing memory access and computational overhead. Since activation values determine which parts of the weight matrix are used, eliminating redundant activations directly minimizes weight matrix accesses, making it particularly effective in memory-bound inference scenarios like auto-regressive LLM decoding. Unlike weight sparsity, activation sparsity does not require structured pruning patterns, allowing for greater flexibility in optimizing execution efficiency.

Deja Vu (Liu et al., 2023) is an early work that utilized activation sparsity in ReLU-based LLMs. It introduced a predictor mechanism to estimate which activations could be set to zero at inference time to reduce computational cost. However, Deja Vu is restricted to ReLU-based models, as ReLU naturally enforces activation sparsity. Since modern LLMs predominantly use SiLU (Shazeer, 2020) activations, alternative methods that directly enable activation sparsity in SiLU-based models are necessary.

To extend activation sparsity to SiLU-based LLMs, researchers proposed methods like ReLUfication Mirzadeh et al. (2023) and ProSparse (Song et al., 2024). These methods replace SiLU or GeLU activations with ReLU, leveraging ReLU's intrinsic sparsity, followed by finetuning to recover model accuracy. These methods require significant computational resources and fine-tuning expertise, making them less practical for large-scale LLMs.

CATS Lee et al. (2024) introduced magnitude-based thresholding to prune SiLU activations, achieving sparsity without retraining. But CATS only exploded the sparsity in the SiLU activations, limiting its sparsity potential. Beyond 50% sparsity, the model performance begins to degrade significantly, restricting its effectiveness. TEAL (Liu et al., 2024) extended activation sparsity by applying thresholding to multiple hidden states in attention and MLP module, achieving about 40% model-wide sparsity. However, TEAL enforces sparsity across all matrices, but not all matrices must be sparsified for optimal acceleration. We identify a better sparsification target, achieving the best trade-off between sparsity and model performance . We also demonstrate that sparsity distribution

is more critical than overall sparsity, allowing REACT to leverage MLP sparsity more efficiently than TEAL, leading to greater actual speedup.

Existing activation sparsity methods either rely on finetuning (ReLUfication, ProSparse) or suboptimal sparsification strategies (CATS, TEAL). Our method, REACT, improves upon these prior works by introducing a better sparsification target, demonstrating that sparsity distribution plays a more critical role than absolute sparsity levels, and designing a custom GPU kernel to fully leverage the discovered sparsity. These advancements enable higher speedup with less accuracy degradation, making activation sparsity a more practical solution for SiLU-based LLM inference.

## 3 BACKGROUND

### 3.1 ACTIVATION SPARSITY AND NOTATIONS

In LLMs, activation sparsity can be further categorized into input sparsity and output sparsity (Song et al., 2024). For a sparse matrix-vector multiplication $\mathbf{y} = \mathbf{x}\mathbf{W}^T$, where $\mathbf{x} \in \mathbb{R}^h, \mathbf{W} \in \mathbb{R}^{h \times d}, \mathbf{y} \in \mathbb{R}^d$, input sparsity occurs when certain elements in the input vector $\mathbf{x}$ are zero, allowing the corresponding columns of the weight matrix $\mathbf{W}$ to be skipped during computation. On the other hand, output sparsity arises when elements in the output vector $\mathbf{y}$ are known to be zero, meaning that the corresponding rows of the weight matrix $\mathbf{W}$ do not need to be loaded. By leveraging activation sparsity in either form, models can significantly reduce memory traffic and computational cost, leading to faster inference.

Additionally, for clarity in the following discussion, we assign specific names to hidden states based on their locations within the MLP computation. For example, we use $\mathbf{W}_{\text{up}}\_out$ to donate the output of the matrix-vector multiplication of $\mathbf{x}$ and $\mathbf{W}_{\text{up}}$, and SiLU$\_out$ to represent the output of the SiLU activation function.

Prior work (Liu et al., 2023) has shown that LLM decoding is memory-bound, with much of the latency stemming from frequent weight matrix transfers rather than computation. This issue is particularly severe for the MLP module, which contains nearly two-thirds of the model parameters, yet remains far less optimized than self-attention. To address this gap, our work focuses on leveraging MLP sparsity for efficient inference.

## 4 REACT: RETHINKING ACTIVATION SPARSITY FOR FASTER LLM DECODING

We propose REACT, a training-free method which extends activation sparsity to modern SiLU-based LLMs. Our goal is to maximize the overall sparsity of the MLP module while ensuring that model accuracy remains above a specified threshold $P\%$. Unlike prior works that only sparsify the activation, REACT explores sparsity in the entire MLP module, allowing sparsification at different hidden stages.

### 4.1 MLP COMPUTATION AND SPARSITY FORMULATION

Given an input $\mathbf{x}$, the standard MLP computation in modern LLMs is expressed as:

$$\text{MLP}(\mathbf{x}) = (\text{SiLU}(\mathbf{x}\mathbf{W}_{\text{gate}}) \odot (\mathbf{x}\mathbf{W}_{\text{up}}))\mathbf{W}_{\text{down}} \tag{1}$$

where $\mathbf{W}_{\text{gate}}, \mathbf{W}_{\text{up}}, \mathbf{W}_{\text{down}}$ are the MLP parameters, SiLU$(\cdot)$ denotes the SiLU activation function, and $\odot$ represents element-wise multiplication.

Within this structure, we identify four possible sparsification positions, as shown in Figure 1a:

$$\begin{aligned} \mathbf{h}_1 &= \mathbf{x}\mathbf{W}_{\text{gate}}, \\ \mathbf{h}_2 &= \text{SiLU}(\mathbf{h}_1), \\ \mathbf{h}_3 &= \mathbf{x}\mathbf{W}_{\text{up}}, \\ \mathbf{h}_4 &= \mathbf{h}_2 \odot \mathbf{h}_3 \end{aligned} \tag{2}$$

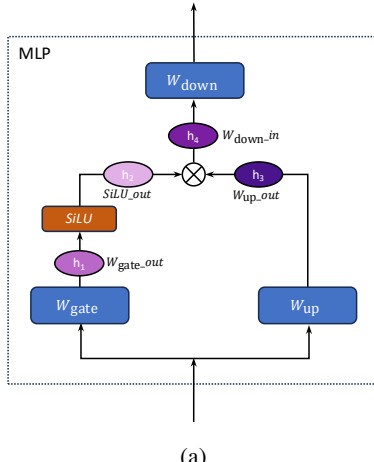
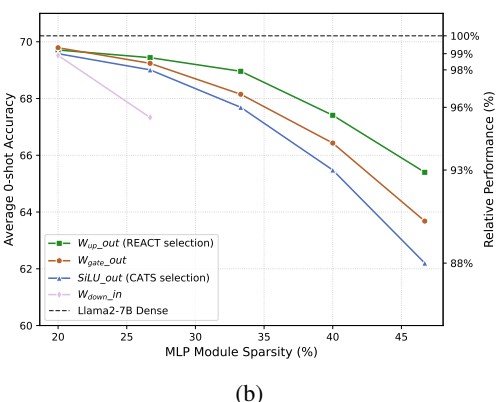

(a)                                              (b)

Figure 1: Left: Illustration of the MLP module structure and sparsification positions in REACT. The four potential sparsification locations are highlighted: $\mathbf{W}_{\text{gate\_out}}$ ( $\mathbf{h}_1$ ), SiLU_out ( $\mathbf{h}_2$ ), $\mathbf{W}_{\text{up\_out}}$ ( $\mathbf{h}_3$ ), and $\mathbf{W}_{\text{down\_in}}$ ( $\mathbf{h}_4$ ). REACT applies activation sparsity at the most effective position, $\mathbf{W}_{\text{up\_out}}$ ( $\mathbf{h}_3$ ), achieving a better trade-off between model accuracy and inference speed. Right: Comparison of different sparsification positions in the MLP module of LLaMA2-7B. The x-axis represents MLP module sparsity, while the left y-axis denotes the average zero-shot accuracy across multiple NLP tasks. The right y-axis shows the relative performance as a percentage of the dense model. The dashed line represents the dense model performance for reference.

The choice of which hidden state $\mathbf{h}_s$ to sparsify directly impacts both computation cost and model accuracy.

## 4.2 MLP SPARSIFICATION OPERATOR

We define a sparsification operator $\mathcal{S}(\mathbf{h}_s, p)$, which applies magnitude-based pruning to the selected hidden state $\mathbf{h}_s$, preserving only the top $(1-p)\%$ largest magnitude elements:

$$\mathbf{h}'_s = \mathcal{S}(\mathbf{h}_s, p), s \in \{1, 2, 3, 4\}. \tag{3}$$

Here, $p$ denotes the pruning ratio, constrained as $0 < p < 100$. In practice, following prior work (Liu et al., 2024), we employ an offline calibration strategy to determine a pruning threshold $t_p$ corresponding to a given sparsity ratio $p$, using a calibration dataset such as WikiText (Merity et al., 2016). Empirically, this method introduces negligible accuracy degradation, making it a practical choice for activation sparsity implementation.

Applying this operator, the sparsified MLP computation is:

$$\text{MLP}_{\text{sparse}}(\mathbf{x}, s, p) = \mathcal{F}(\mathcal{S}(\mathbf{h}_s, p)) \tag{4}$$

where $\mathcal{F}(\cdot)$ represents the MLP forward computation, which adapts based on the sparsified hidden state:

$$\mathcal{F}(\mathbf{h}'_s) = \begin{cases} (\text{SiLU}(\mathbf{h}'_s) \odot \mathbf{h}_3)\mathbf{W}_{\text{down}}, & s = 1, 2 \\ (\text{SiLU}(\mathbf{h}_1) \odot \mathbf{h}'_s)\mathbf{W}_{\text{down}}, & s = 3 \\ (\text{SiLU}(\mathbf{h}_1) \odot \mathbf{h}_3)\mathbf{W}'_{\text{down}}, & s = 4 \end{cases} \tag{5}$$

Here, $\mathbf{W}'_{\text{down}}$ accounts for cases where sparsification is applied to its input.

We define the global sparsity of the MLP module as the average sparsity of its three weight matrices: $\mathbf{W}_{\text{gate}}, \mathbf{W}_{\text{up}}$, and $\mathbf{W}_{\text{down}}$. The sparsity of a weight matrix is measured as the fraction of its elements that are zero.

---

**Algorithm 1: Sparse MLP Fused Kernel**

**Input:** $W_{\text{gate}}, W_{\text{up}}, W_{\text{down}}, x, t$

1   $x \leftarrow \textbf{Load}(x), W_{\text{up}} \leftarrow \textbf{Load}(W_{\text{up}})$
2   $h_3 \leftarrow xW_{up}$
3   $mask_1 \leftarrow (|h_3| \geq t)$
4   $W'_{\text{gate}} \leftarrow \textbf{Load}(W_{\text{gate}}, \text{mask} = mask_1)$
5   $h_4 \leftarrow \text{SiLU}(xW'_{\text{gate}}) * h_3$
6   $mask_2 \leftarrow (h_4 \neq 0)$
7   $W'_{\text{down}} \leftarrow \textbf{Load}(W_{\text{down}}, \text{mask} = mask_2)$
8   $y \leftarrow \textbf{Store}(h_4 W'_{\text{down}})$

---

### 4.3 OPTIMIZATION OBJECTIVE

REACT aims to maximize the overall sparsity of the MLP module while maintaining model accuracy above $P\%$ as follows:

$$\max_{s,p} S_{\text{MLP}}, \text{ subject to } \mathcal{A}_{\text{sparse}} \geq P\%. \tag{6}$$

where $\mathcal{A}_{\text{sparse}}$ represents the mean performance across multiple downstream NLP tasks. Typically, we set $P \geq 96\%$ to ensure no significant degradation in model performance.

Since $s$ remains a free variable in our method, we empirically evaluate different sparsification locations, i.e., $\mathbf{h}_1, \mathbf{h}_2, \mathbf{h}_3, \mathbf{h}_4,$, corresponding to the four candidate hidden states within the MLP module. Through extensive experiments, we determine the optimal choice $s^*$ that maximizes MLP sparsity without compromising model accuracy. This selection is not predefined but emerges from empirical evaluation, allowing REACT to be adaptable across different LLM architectures.

### 4.4 SPARSITY-AWARE CUSTOM GPU KERNELS

To achieve wall-clock speedup in LLM decoding, we develop a highly efficient GPU kernel that fully exploits activation sparsity. Given that LLM decoding is a memory-bound process, our design prioritizes minimizing unnecessary memory accesses, particularly avoiding redundant loads and stores of intermediate states by keeping them in registers or shared memory. Additionally, we ensure coalesced memory access, which significantly improves global memory bandwidth utilization.

Based on these design principles, we propose a sparse MLP fused kernel, outlined in Algorithm 1, which provides a memory-centric perspective on the computation. Importantly, our design remains generalizable: regardless of the sparsification position $s$, the kernel structure remains largely similar. Therefore, for clarity, we present the case where $s = 3$ as a representative example. In Alg. 1, all memory load and store operations are explicitly highlighted.

Compared to previous approaches, our kernel eliminates all redundant loads and stores of intermediate states, ensuring minimal memory access overhead. To achieve this, we implement the entire sparse MLP computation within a single fused kernel, preventing unnecessary global memory operations. A natural consequence of this design is the reduction in kernel launch overhead, further improving efficiency.

Additionally, we introduce a strided row-major storage format to improve weight matrix access efficiency. Unlike TEAL, which stored weights in column-major format, we store all three weight matrices in a row-major format, but with $\mathbf{W}_{\text{down}}$ transposed, and stored in a contiguous memory layout to unify access strides across all matrices. This design improves data locality and ensures coalesced memory access, significantly enhancing memory throughput.

## 5 EXPERIMENTS

In this section, we comprehensively evaluate REACT through a series of experiments designed to analyze its effectiveness in LLM decoding. We first determine the optimal sparsification position in LLaMA2-7B and validate this choice across other models, including the LLaMA family and Mistral. Next, we investigate how different sparsity distributions affect speedup by benchmarking single

|            | **Dense** | $\mathbf{W_{up}}\_out$ | $\mathbf{W_{gate}}\_out$ | **SiLU**$\_out$ |
|------------|-----------|------------------------|--------------------------|-----------------|
| Perplexity | 4.95      | 5.47                   | 5.72                     | 5.82            |

Table 1: Perplexity results on the WikiText dataset at 40% sparsity across different sparsification positions.

MLP module inference times. We further conduct micro-benchmarks to quantify the contribution of each kernel-level optimization. Finally, we integrate REACT into LLMs and measure end-to-end decoding latency to demonstrate it's wall-clock speedup for LLM decoding. To ensure reproducibility, we provide the code in the supplementary materials.

## 5.1 OPTIMAL SPARSIFICATION POSITION SELECTION

To determine the optimal sparsification position within the MLP module, we first conduct evaluations on LLaMA2-7B Touvron et al. (2023) and validate our findings across different models in the next section. We follow prior work and utilize Eleuther AI's LM Evaluation Harness (Gao et al., 2024) to assess model performance via average zero-shot accuracy across a diverse set of NLP tasks. These tasks encompass three main categories: commonsense reasoning (ARC-Easy, ARC-Challenge (Clark et al., 2018), HellaSwag (Zellers et al., 2019), OpenBookQA (Mihaylov et al., 2018), Winogrande (Sakaguchi et al., 2021)), reading comprehension (BoolQ (Clark et al., 2019), SciQ (Welbl et al., 2017)), and language modeling (LAMBADA) (Paperno et al., 2016). Additionally, we evaluate language modeling performance using perplexity on the WikiText (Merity et al., 2016) validation set, with a context size of 2048 and an evaluation window of 512.

For consistency, all sparsity levels in the following refer to MLP module-level sparsity, i.e., the average sparsity across the three MLP weight matrices.

**Results.** Figure 1b presents the model's performance across different MLP sparsity levels for all eight evaluation tasks. At approximately 27% sparsity, all positions except $\mathbf{W_{down}}\_in$ maintain at least 98% of the original model's accuracy. The weaker performance of $\mathbf{W_{down}}\_in$ stems from its limited impact on overall MLP sparsity. Since this position only sparsifies $\mathbf{W_{down}}$, achieving 27% overall MLP sparsity translates to an extreme 80% sparsity in $\mathbf{W_{down}}$ alone. In contrast, other positions distribute sparsity across two matrices, maintaining a more moderate 40% sparsity per matrix, which helps preserve model performance.

As sparsity increases to 40%, only $\mathbf{W_{up}}\_out$ retains over 96% accuracy, while SiLU$\_out$, the position used in CATS, drops to about 93%, leading to a significant degradation in model performance. Beyond 47% sparsity, all positions exhibit substantial accuracy loss, surpassing our predefined threshold of $P \geq 96\%$ in the section 4.

Table 1 further confirms this trend by reporting perplexity results on the WikiText dataset at 40% sparsity across different sparsification positions. Here, $\mathbf{W_{up}}\_out$ achieves the lowest perplexity, outperforming SiLU$\_out$, further justifying our choice.

These findings indicate that $\mathbf{W_{up}}\_out$ is the most effective sparsification position, as it maintains high accuracy and achieves the lowest perplexity under increasing sparsity constraints. In the next section, we extend our evaluation to a broader range of models to verify the generalizability of this choice.

## 5.2 GENERALIZATION ACROSS DIFFERENT LLMs

To verify that $\mathbf{W_{up}}\_out$ remains the optimal sparsification position, we extend our evaluation to multiple models, including LLaMA2-7B, LLaMA2-13B, LLaMA3-8B (Dubey et al., 2024), and Mistral-7B (Jiang et al., 2023). Across different models and sparsity levels, we demonstrate that REACT consistently outperforms CATS in preserving model accuracy.

We evaluate all methods at sparsity levels of 20%, 30%, and 40%. In REACT, we further test only three positions: $\mathbf{W_{up}}\_out$, $\mathbf{W_{gate}}\_out$, and SiLU$\_out$, as $\mathbf{W_{down}}\_in$ provides limited sparsity (max 33%) and was shown in the previous experiment to degrade model performance. Notably, SiLU$\_out$ corresponds to the setting used in CATS. For a fair comparison, we evaluate TEAL using its uniform

Table 2: Average zero-shot accuracy across different models and sparsity levels.

| Method | Sparse Position | LLaMA2-7B | LLaMA2-13B | LLaMA3-8B | Mistral-7B |
|---|---|---|---|---|---|
| **Dense Model** | - | 70.21 | 72.78 | 73.99 | 74.19 |
| CATS 20% | $SiLU\_out$ | 69.58 | 72.56 | 73.31 | 73.68 |
| REACT 20% | $\mathbf{W}_{gate}\_out$ | 69.79 | 72.58 | 73.31 | 73.58 |
| REACT 20% | $\mathbf{W}_{up}\_out$ | 69.71 | 72.70 | 73.37 | 73.93 |
| CATS 30% | $SiLU\_out$ | 68.46 | 71.69 | 71.20 | 72.60 |
| REACT 30% | $\mathbf{W}_{gate}\_out$ | 68.85 | 71.68 | 70.90 | 72.68 |
| REACT 30% | $\mathbf{W}_{up}\_out$ | 69.00 | 72.38 | 72.23 | 73.96 |
| CATS 40% | $SiLU\_out$ | 65.48 | 69.46 | 65.09 | 68.14 |
| REACT 40% | $\mathbf{W}_{gate}\_out$ | 66.43 | 70.04 | 65.21 | 70.46 |
| REACT 40% | $\mathbf{W}_{up}\_out$ | 67.41 | 71.13 | 68.31 | 72.27 |
| TEAL 55% | $\mathbf{x}, \mathbf{W}_{down}\_in$ | 66.94 | 71.30 | 69.38 | 72.27 |

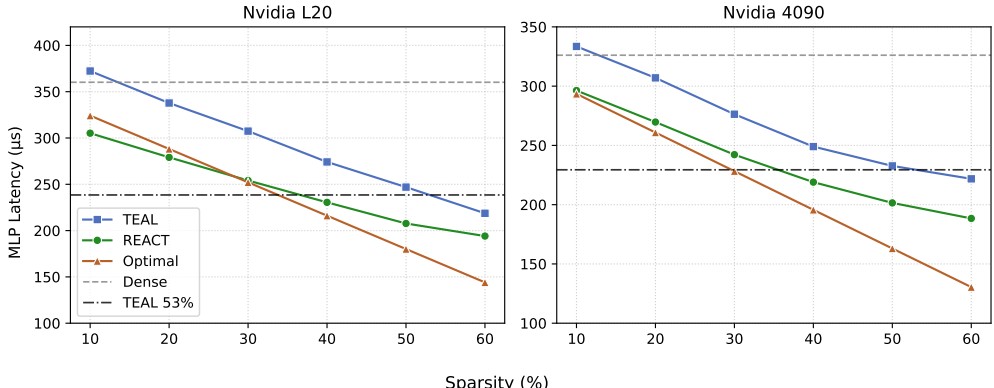

Figure 2: MLP latency of different sparsity levels on Nvidia L20 and RTX 4090 GPUs. The x-axis represents the overall sparsity of the MLP module, while the y-axis denotes the measured latency (μs). The dashed black line indicates TEAL's latency at 53% sparsity, corresponding to the model's 96% accuracy constraint.

sparsification variant, applying sparsity only to the MLP module while keeping the attention layers dense.

**Results.** Table 2 presents the average zero-shot accuracy across eight evaluation tasks for different models. At 40% sparsity, REACT maintains over 96% of the original model's accuracy in LLaMA2-7B, LLaMA2-13B, and Mistral-8B. Notably, in Mistral-8B, REACT outperforms CATS by 5.57%, achieving 97.41% of the dense model's accuracy, indicating that Mistral exhibits the highest sparsity tolerance, potentially due to its larger MLP capacity.

However, we observe slightly higher accuracy degradation in LLaMA3-8B, aligning with similar findings in TEAL (Liu et al., 2024). A possible explanation is differences in activation distributions or pretraining regularization strategies, which may affect sparsity retention. While a more detailed investigation is beyond the scope of this paper, we leave this as an open direction for future study.

For TEAL, we report its sparsity at 96% accuracy retention, which reaches near 55% MLP sparsity. While TEAL achieves slightly higher sparsity than REACT at the same accuracy target, this does not necessarily imply greater speedup. In the next section, we demonstrate that the distribution of sparsity across MLP sub-components plays a critical role in inference acceleration.

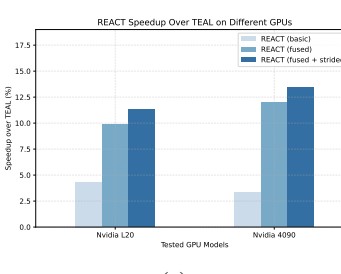
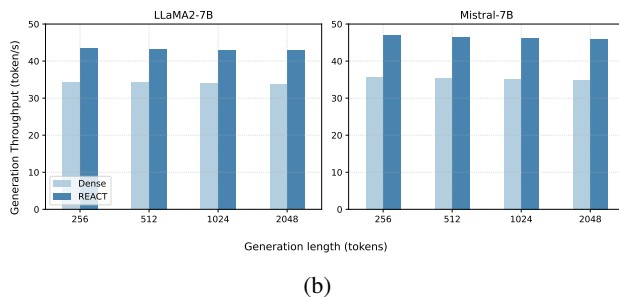

|     |     |
| :-: | :-: |
| (a) | (b) |

Figure 3: Left: Speedup of REACT over TEAL on different GPUs. The y-axis shows the percentage latency reduction of three REACT kernel variants compared to TEAL. Right: Generation throughput of different generation length on LLaMA2-7B and Mistral-7B models. The x-axis represents different generation lengths, while the y-axis denotes the generation throughput measured in token/s.

## 5.3 Impact of Sparsity Distribution on Inference Efficiency

We conduct all experiments on NVIDIA L20 and RTX 4090 GPUs, with memory bandwidths of 768GB/s and 1008GB/s, respectively. These GPUs are selected to ensure a broader evaluation of our sparsity-aware optimizations across different memory bandwidth GPUs. We use PyTorch v2.4.0, CUDA v12.2, and HuggingFace Transformers v4.44.2 to implement dense LLM inference. All models and micro-benchmarks run in FP16 to fit within the 24GB memory limit for sub-8B models. Our sparsity-aware custom GPU kernels are implemented using Triton v3.0.0. To ensure robust timing measurements, we report latency as the geometric mean over multiple runs.

We evaluate the MLP module latency in a single decoder layer of LLaMA2-7B, comparing different sparsity distributions at the same overall MLP sparsity level. For a fair comparison, REACT adopts the same optimizations as TEAL and does not include additional enhancements introduced in Section 4.4.

**Results.** As shown in Figure 2, kernel runtime decreases approximately linearly as sparsity increases, but beyond 50% sparsity, we observe diminishing returns due to kernel launch overheads and other fixed computational costs.

Another clear trend is that at the same overall sparsity, REACT consistently achieves lower MLP module latency than TEAL. This is because the speedup obtained from sparsity is not solely determined by the overall sparsity ratio but also by how sparsity is distributed across weight matrices. REACT's sparsity pattern ($\mathbf{W}_{\text{up}}$: 0%, $\mathbf{W}_{\text{gate}}$: 75%, $\mathbf{W}_{\text{down}}$: 75%) outperforms TEAL's uniform sparsity distribution in reducing latency. Additionally, we evaluated TEAL's greedy search algorithm with various sparsity distributions on Nvidia 4090 and found that in all cases, the resulting latency remained similar to TEAL's uniformly sparsity allocation.

Finally, we address the question raised in the previous section: while TEAL achieves slightly larger overall sparsity at the same model accuracy, how does this impact inference speedup? In Figure 2, we compare the sparsity levels of REACT and TEAL at the same MLP module latency.

We observe two key facts: (1) At the latency where TEAL reaches 53% sparsity, which corresponds to the threshold at which TEAL maintains 96% of the original LLaMA2-7B model's accuracy, REACT achieves the same speed with only 35% overall sparsity on both GPUs. (2) REACT's module latency continues to decrease as sparsity increases.

These observations lead to a critical conclusion: since REACT at 35% sparsity already matches TEAL's latency at 53% sparsity, and REACT at 40% sparsity achieves even lower latency, it follows that REACT attains a higher speedup than TEAL under the same model accuracy constraints. This highlights that sparsity distribution plays a more crucial role in acceleration than overall sparsity percentage alone.

## 5.4 KERNEL MICRO-BENCHMARK

In this section, we evaluate the two key contributions of our sparsity-aware custom GPU kernel introduced in section 4.4. We conduct experiments on L20 and RTX 4090 GPUs, using the same MLP module setup as in the previous section. To ensure a fair comparison, we measure the latency of both methods while maintaining 96% of the original model's accuracy. Under this constraint, TEAL achieves 53% sparsity, whereas REACT requires only 40% sparsity.

We report the percentage reduction in latency relative to TEAL for three versions of our GPU kernel, as shown in Figure 3a. Although REACT's overall MLP sparsity is slightly lower than TEAL (40% vs. 53%), its better sparsity distribution enables significantly lower latency. This further validates our hypothesis that the distribution of sparsity across MLP submodules has a greater impact on inference speed than the overall sparsity ratio.

For our two kernel optimizations, we observe the following improvements: (1) The fused kernel reduces latency by approximately 7% compared to the basic kernel, primarily by eliminating redundant memory load/store operations and reducing kernel launch overhead. (2) Adopting our strided row-major storage format improves data locality and ensures coalesced memory access, leading to an additional 1.4% latency reduction over the fused kernel.

Overall, our sparsity-aware custom GPU kernel achieves up to 13.4% lower latency than TEAL, while maintaining the same model accuracy. In the next section, we evaluate end-to-end LLM decoding performance using REACT.

## 5.5 END-TO-END LLM DECODING PERFORMANCE

In this section, we integrate our custom GPU kernel into LLaMA2-7B and Mistral-7B and evaluate their decoding throughput at different generation lengths. To ensure no significant degradation in model performance ($P \geq 96\%$), we set MLP sparsity to 40% for LLaMA2-7B and 41% for Mistral-7B, reflecting Mistral's higher sparsity tolerance observed in previous experiments. We measure throughput in tokens per second (TPS), computed as output length divided by total decoding latency.

As shown in Figure 3b, REACT consistently accelerates LLM decoding across different generation lengths, achieving **1.26×** and **1.33×** speedup for LLaMA2-7B and Mistral-7B, respectively. This improvement demonstrates that our sparsity-aware kernel effectively translates MLP sparsity into actual inference speedup, even without modifying attention layers.

While TEAL achieves higher throughput by sparsifying both attention and MLP layers, REACT demonstrates superior speedup at the same model accuracy, achieving a 13.4% latency reduction over TEAL in MLP module as shown in section 5.4 due to its better sparsity allocation and optimized kernel design.

## 6 CONCLUSION

We propose REACT, a training-free activation sparsity method that accelerates LLM inference by leveraging activation sparsity within the MLP module, while optimizing its distribution to further enhance efficiency. Through systematic experiments, we validate that a well-balanced sparsity distribution achieves higher speedup with the same model performance. On LLaMA2-7B and Mistral-7B, REACT delivers 1.26× and 1.33× speedup, respectively, with minimal accuracy loss. These improvements make REACT particularly valuable for deploying LLMs on resource-constrained environments, where efficient decoding is critical for real-time applications.

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

# A    APPENDIX

## A.1    THE USE OF LARGE LANGUAGE MODELS (LLMS)

We only use AI tools like ChatGPT to polish the written text.

