# OpenReview forum: "Sparsity Distribution Matters: REACT for Accelerating Large Language Models"
_ICLR.cc/2026/Conference — Submitted to ICLR 2026_

### Official Review · Reviewer_R9xS · 2025-10-24

**Soundness:** 2
**Presentation:** 2
**Contribution:** 2
**Rating:** 2
**Confidence:** 4

**Summary:**

The paper introduces REACT,  a method that utilizes the activation sparsity of the output of up projection to speed up the inference of LLMs. REACT proposes to mask out the output of up projections via its magnitude to reduce the I/O and computation of down projection and gate projection. The paper evlauates REACT on various benchmarks for LLaMA2-7B, LLaMA2-13B, LLaMA3-8B, Mistral-7B.

**Strengths:**

The paper conduct experiments on various LLMs, including LLaMA2-7B, LLaMA2-13B, LLaMA3-8B and Mistral-7B.

**Weaknesses:**

1. Table 2 presents a comparison between the proposed REACT and TEAL methods. However, I do not observe any clear advantage of REACT over TEAL. In fact, TEAL with 55% activation sparsity outperforms REACT with 40% activation sparsity on multiple models, including LLaMA2-13B, LLaMA3-8B, and Mistral-7B.

2. Figure 2 indicates that REACT achieves faster inference speed compared to TEAL, which seems counterintuitive. REACT applies activation sparsification only to the down-projection and gate-projection layers, while TEAL applies it to all projection layers (down/up/gate). Could the authors provide more details to clarify why REACT surpasses TEAL in inference speed under the same activation sparsity level?

3. The authors only report average zero-shot accuracy. It is recommended to include detailed per-task results in the appendix to better understand performance variations across benchmarks.

4. The current evaluation does not cover challenging reasoning, mathematics, and code generation tasks such as MMLU, GSM-8K, MATH, and HumanEval. For perplexity evaluation, it would be more appropriate to use C4 instead of WikiText, as C4 is much larger in scale. Furthermore, the paper omits experiments on Qwen-2.5/3 models, which generally outperform LLaMA-2/3 and Mistral and may exhibit different sensitivities to activation sparsification.

**Questions:**

see Weaknesses

---

### Official Review · Reviewer_rmuT · 2025-10-31

**Soundness:** 2
**Presentation:** 2
**Contribution:** 2
**Rating:** 2
**Confidence:** 4

**Summary:**

This work proposes REACT, a training free sparsification method that accelerates inference through sparsification of MLP layers. Contributions include determining locations for sparsifications, custom GPU kernels for inference and achieving end-to-end speedup for LLaMA2-7B and Mistral-7B models.

**Strengths:**

- Authors systematically explore potential regions for sparsifications in the MLP module. Analysis in Figure(1) and Table(1) clearly demonstrates a target for sparsification useful for future research.
- Theoretical  sparsity is realized practically through a sparsity-aware GPU kernel. Provided results establish activation sparsity with speedups without retraining the model.

**Weaknesses:**

**Unclear contribution/comparison w.r.t prior works.**
-   CATS (Lee et al 24) uses magnitude based pruning on SiLU_out(h_2), while TEAL (Liu et al 24) increments the sparsity levels of W_up, W_down and W_gate separately which allows for converging to the optimal sparsity distribution across the three FFN weights. The additional contribution of REACT is hence unclear.
	-  By comparing with TEAL at uniform distribution to all weights and not using the layer-wise greedy approach, the performance of TEAL will be hit considerably. The paper should expand on the comparison with TEAL’s greedy search algorithm and their converged sparsity levels.
- There are three variables, (1) inference speedup, (2) sparsity ratio, and (3) downstream performance.
	-   With fixed (2): REACT > CATS in (3) , what about (1) ? What are the differences in speedup on sparsifying different weight matrices?
	-   For TEAL, authors fix (3) and compare (1). How does (1) vary with different (2) here?
	-   How much does the implementation affect the speedup here?
	-   The text presents the relation between the three variables for the three methods very vaguely.
	-   Authors should present 2D plots relating first variable with the second, while fixing third. For all three methods.

**Need More details on Method/Evals**
-   It is unclear how authors solve equation (6). If it is through extensive experiments (line 238), how were the various variables (layers, what sparsifications to test, which datasets to evaluate) determined?
-   The metric used to compare various baselines, “average zero-shot accuracy” is used without any regard to variance between evals. Considering that most numbers are really close, I would suggest providing a more convincing argument for improvement in performance.

**Need more details on “Sparsity-Aware GPU kernel”**
-   "(line 444)...The fused kernel reduces latency by approximately 7% compared …”
	-   Sparsity aware fused GPU kernels are not new and several works including CATS have provided them before. Consider outlining the new contributions.
-   "(line 259) Strided row-major format improves data locality and ensures coalesced memory access.. leading to additional 1.4% latency reduction.."
    -   Improvement in memory coalescing across multiple threads due to a specific storage format is very well known. If the storage format matches the memory access patterns, you will always see a slight improvement. Hence it is possible to change the implementation to change the optimal memory storage. Further, "1.4%" seems marginal without further details.
- If  authors can explain how their kernel is new and different from existing implementations, and further provide reproducible benchmarking information, it would be helpful.

**Minor Presentation Improvement needed**
- Section 5.1 on Page 6 discusses Figure 1(b) on Page 4.
- The content of Section 4.4 helps in understanding Section 5.4. Consider getting them closer or a single section.
- I do not fully understand Equation (5), are you trying to say:

**s=1:**  (SiLU(S(h_1, p)) ⊙ h_3) W_down

**s=2:** (S(SiLU(h_1), p) ⊙ h_3) W_down

**s=3:** (SiLU(h_1) ⊙ S(h_3, p)) W_down

**s=4:** S(SiLU(h_1) ⊙ h_3, p) W_down

**Questions:**

- Regarding CATS (Lee et al 24), authors claim that "..beyond 50% sparsity, the model performance begins to degrade significantly, restricting its effectiveness.", isn’t the same true for REACT?

---

### Official Review · Reviewer_R26R · 2025-10-31

**Soundness:** 2
**Presentation:** 2
**Contribution:** 2
**Rating:** 4
**Confidence:** 3

**Summary:**

This paper proposes REACT,  a training-free sparsification method that optimizes sparsity distribution within the Multi-Layer Perceptron (MLP) module, improving inference speed without sacrificing model performance.

**Strengths:**

1. The paper writing is clear with good presentation.
2. The results are good.

**Weaknesses:**

1. It would be better if the authors could showcase the results on larger models, such as 13B or 30B.
2. It would be better if the authors could evaluate on datasets like LongBench and RULER.

**Questions:**

Please see the weaknesses.

---

### Official Review · Reviewer_Qfzw · 2025-11-03

**Soundness:** 3
**Presentation:** 3
**Contribution:** 3
**Rating:** 4
**Confidence:** 3

**Summary:**

This paper propose REACT as a training-free sparification method for MLP module with a focus on sparsity distribution. Experimental results demonstrate that the quality can be maintained while improving inference speed.

**Strengths:**

This paper addresses an important question, and the authors have made a commendable effort to obtain practical benefits through custom GPU kernels. I like that the authors take practical acceleration into account when designing the algorithm.

**Weaknesses:**

- My main question is, what exactly is the proposed method? Am I correct in thinking it just involves sparsifying the MLP layer?
- Also, while implementing this algorithm in the kernel is impressive, what makes the kernel approach different from previous methods? Is it the layout transpose?
- Am I correct in understanding that the proposed methods are better than comparisons only for LLaMA-2 7B at a given sparsity ratio (if not taking into account the practical speedups for now)?

**Questions:**

- I don't quite understand why the paper didn't mention experiment results on LLaMA-3 when there are experiments on it?
- Could the authors clarify the setup used for measuring end-to-end inference latency? Specifically, is the measurement conducted with vLLM, SGLang, or HuggingFace?
- Line 235, where does the number `P ≥ 96%` come from?
- Line 238, could the authors elaborate on the "empirical evaluation" to get `s*`?

---

### Meta-Review · Area_Chair_zCrF · 2026-01-05

**Summary:**

Strength: The paper considers an important problem. The writing of the paper is good, and the experimental results are positive across various LLMs.

Weakness: The novelty of the proposed method needs better explanation, especially how it improves upon prior approaches should be clearly explained. The experimental section should be improved by providing more detailed explanation and including more tasks.

**Reviewer Concerns:**

Weakness: The novelty of the proposed method needs better explanation, especially how it improves upon prior approaches should be clearly explained. The experimental section should be improved by providing more detailed explanation and including more tasks.

**Reviewer Scores:**

The authors did not submit any rebuttal response. Thus, the reviewers will likely keep their scores.

---

### Decision · Program_Chairs · 2026-01-26

Reject